# Cell-Specific Immune Regulation by Glucocorticoids in Murine Models of Infection and Inflammation

**DOI:** 10.3390/cells11142126

**Published:** 2022-07-06

**Authors:** Lourdes Rocamora-Reverte, Andreas Villunger, G. Jan Wiegers

**Affiliations:** 1Department of Immunology, Institute for Biomedical Aging Research, University of Innsbruck, 6020 Innsbruck, Austria; lourdes.rocamora@uibk.ac.at; 2Institute for Developmental Immunology, Biocenter, Medical University of Innsbruck, 6020 Innsbruck, Austria; andreas.villunger@i-med.ac.at

**Keywords:** glucocorticoid, glucocorticoid receptor, T cell, regulatory T cell, innate lymphoid cell, myeloid cell, macrophage, B cell

## Abstract

Glucocorticoids (GC) are highly potent negative regulators of immune and inflammatory responses. Effects of GC are primarily mediated by the glucocorticoid receptor (GR) which is expressed by all cell types of the immune system. It is, therefore, difficult to elucidate how endogenous GC mediate their effects on immune responses that involve multiple cellular interactions between various immune cell subsets. This review focuses on endogenous GC targeting specific cells of the immune system in various animal models of infection and inflammation. Without the timed release of these hormones, animals infected with various microbes or challenged in inflammatory disease models succumb as a consequence of overshooting immune and inflammatory responses. A clearer picture is emerging that endogenous GC thereby act in a cell-specific and disease model-dependent manner, justifying the need to develop techniques that target GC to individual immune cell types for improved clinical application.

## 1. Introduction

The potency of glucocorticoids (GCs) as negative regulators of immune cells and inflammatory effector molecules is widely accepted and as such they are successfully used in the treatment of autoimmune and inflammatory diseases [1,2]. Glucocorticoid biosynthesis occurs in the adrenal cortex, although there is growing evidence supporting the significance of extra-adrenal GC synthesis. Thus, locally synthesized GCs adjust T cell selection in the thymus [3,4,5] and help in the control of inflammatory processes in the epithelial barriers of the skin, lung, and intestine [6,7]. In addition, GCs can be regenerated from inactive precursors in cells expressing the enzyme 11β-hydroxysteroid dehydrogenase type 1 (Hsd11b1) [5,8]. GCs primarily act via the glucocorticoid receptor (GR), also known as NR3C1, a member of the nuclear receptor family, which is present in the cytoplasm in a multiprotein complex that contains heat-shock proteins (Hsp70, Hsp90), immunophilins and other chaperones. The molecular mechanisms of GC action are manifold and, under some conditions, they may include the regulation of as much as 20% of the genome (for review, see [2]). In addition to this diversity in gene regulation, endogenous GC production is under the control of circadian rhythms and accumulated evidence suggests that GCs are key regulators of the diurnal rhythm of activity that has been reported both in innate and adaptive immunity ([9]; for review, see [10]). Because the GR is almost ubiquitously expressed, GCs affect essentially all immune cells, making the interpretation of effects in mice carrying global alterations of the GR challenging. Moreover, the activity of the primary system regulating both the circadian and stress-induced release of GC, the hypothalamus-pituitary-adrenal (HPA) axis, is also changed in these animal models, which causes further difficulties in the evaluation of the data collected. Over the last 20 years, the role of GCs in infectious, autoimmune and inflammatory disease models has been investigated using cell-type specific, conditional GR knockout mice. Interestingly, it has emerged that the inhibitory effects of endogenous GC are mediated in a remarkably cell-type and disease model-dependent manner. An overview of GC effects (both endogenous and exogenous) on various immune cell subsets in murine models of infection and inflammation is the subject of this review.

## 2. GC and Immune Cell Subsets

### 2.1. GC and Innate Lymphoid Cells

Innate lymphoid cells (ILCs) are primarily tissue-resident cells of the innate immune system that are among the first cells to respond to infection by the production of cytokines. They are a heterogeneous group of lymphocytes present in barrier tissues of the host (e.g., gut, lung) displaying tissue-specific functions [11]. The cytokine profile they produce depends on the nature of the microorganism causing infection (e.g., intracellular or extracellular). Acute viral infection, for example, requires the production of interferon-γ (IFN-γ) for antiviral defense and ILCs producing this cytokine include natural killer (NK) cells and group 1 ILC (ILC-1) [11]. Quatrini et al. studied infection with murine cytomegalovirus (MCMV) in mice lacking GR expression in NK and ILC-1 cells (GR^Ncr1-iCre^ mice) and found that IFN-γ production two days after infection was increased in the spleen, but not in the liver (endogenous GCs are released within 36 h in this infection model) [12]. Increased IFN-γ production induced splenic hyper-inflammation and immunopathology, resulting in the reduced survival of GR^Ncr1-iCre^ mice, as compared to their WT littermates. Interestingly, viral clearance was not compromised in GR^Ncr1-iCre^ mice, suggesting that the hyper-production of IFN-γ and downstream signaling were responsible for the increase in mortality [12]. The same mice were analyzed in an experimental model of endotoxin (LPS) tolerance, the establishment of which has been shown to depend on endogenous GCs [13,14]. Indeed, endogenous GC release reportedly inhibits IFN-γ production by NKp46^+^ ILC, thereby allowing for the production of IL-10 by myeloid cells which, in turn, produces an immune suppressive state, effectively tolerizing these mice against LPS. In GR^Ncr1-iCre^ mice, however, higher levels of IFN-γ along with lower IL-10 concentrations produced a loss of tolerance to LPS and strongly reduced survival [14] (see Figure 1 and Table 1).

ILC group 2 (ILC-2) cells are broadly distributed along mucosal surfaces and react in response to epithelial cell stress with cytokine production (such as IL-4, IL-5 and IL-13) to initiate inflammation. Studies by the lab of Cidlowski revealed that endogenous GC were instrumental in preventing the development of spontaneous gastric inflammation and spasmolytic polypeptide-expressing metaplasia (SPEM), the latter being a lesion in the gastric mucosa that is considered to be a putative precursor of gastric cancer in a chronic inflammatory setting [15]. Subsequent work discovered that both GC and androgens target ILC-2 cells that co-express high levels of both the GR and the androgen receptor (AR). Treatment with either GC or androgens suppressed proinflammatory cytokine production by these cells and, strikingly, ILC-2 cell depletion protected adrenalectomized and castrated mice from SPEM development [16].

**Table 1 cells-11-02126-t001:** Immune cell types targeted by GC in animal models of infection, autoimmunity and inflammation.

Animal Model	Cells Targeted by GC	Source of GC	Observed Effects Upon Conditional GR Deletion in Targeted Cells	Reference
MCMV Infection	NK + ILC-1	endogenous	splenic hyper-inflammation, survival ↓	[12]
LPS tolerance	NK + ILC-1	endogenous	loss of LPS tolerance, survival ↓	[14]
Gastric inflammation, SPEM	ILC-2	endogenous	spontaneous gastric inflammation in ♀, protection by GC (and androgens) *	[16]
Polyclonal T cell activation	T cell	endogenous	survival ↓, rescue by COX-2 inhibition	[17]
Cecal ligation and puncture (CLP)	T cell	endogenous	survival ↓	[18]
*Toxoplasma gondii* infection	T cell	endogenous	hyperactive CD4+ T cell response, survival ↓	[19]
Experimental autoimmune encephalomyelitis (EAE)	T cell	endogenous	disease onset earlier, more severe course	[20]
Experimental autoimmune encephalomyelitis (EAE)	T cell	exogenous	resistance to DEX treatment, reduced induction of apoptosis in Th17 cells	[20]
Allergic airway inflammation	T cell	exogenous	no impact on DEX treatment, airway epithelial cells crucial GC target	[21]
Antigen-induced arthritis	T cell	exogenous	resistance to DEX treatment, circulating pro-inflammatory cytokines ↑	[22]
Contact dermatitis	T cell	exogenous	no impact on DEX treatment	[23]
Graft-versus-host disease (GvHD)	T cell	endogenous	strongly aggravated clinical disease, accelerated death	[24]
Experimental autoimmune encephalomyelitis (EAE)	Foxp3^+^ T cell	exogenous	resistance to DEX treatment, impaired Treg cell function	[25]
Allergic airway inflammation	Foxp3^+^ T cell	exogenous	resistance to DEX treatment, lung-infiltrating proinflammatory CD4^+^ T cells ↑	[25]
Experimental Inflammatory Bowel Disease (IBD)	Foxp3^+^ T cell	endogenous	failure to prevent inflammatory bowel disease, loss of Treg cell phenotype	[26])
Contact dermatitis	macrophages, neutrophils	exogenous	resistance to DEX treatment, massive leukocyte infiltration of the skin	[23]
Allergic airway inflammation	macrophages, neutrophils	exogenous	no impact on DEX treatment, airway epithelial cells crucial GC target	[21]
Endotoxaemia	macrophages; neutrophils?	endogenous	increased circulating pro-inflammatory cytokines, survival ↓	[27]
Antigen-induced arthritis	macrophages, neutrophils	exogenous	no impact on DEX treatment	[22]
DSS-induced colitis	macrophages, neutrophils	endogenous	failure to resolve inflammation, increased cytokine expression in colon	[28]
Myocardial infarction	macrophages	endogenous	impaired post-ischemic angiogenesis, reduced cardiac function, survival ↓	[29]
Endotoxaemia	dendritic cell	endogenous	increased circulating pro-inflammatory cytokines, survival ↓	[13]
Allergic airway inflammation	dendritic cell	exogenous	no impact on DEX treatment, airway epithelial cells crucial GC target	[21]
Antigen-induced arthritis	dendritic cell	exogenous	no impact on DEX treatment	[22]
Antigen-induced arthritis	B cell	exogenous	no impact on DEX treatment	[22]
Allergic airway inflammation	B cell	exogenous	no impact on DEX treatment, airway epithelial cells crucial GC target	[21]

* ILC-2 cells were depleted by antibody treatment.

Very little is known about whether endogenous GCs regulate ILC-3 cells, which act constitutively to regulate the balance between the intestinal barrier and commensal microbiota, although it was reported that dexamethasone suppressed IL-23-mediated IL-22 production in human and mouse ILC-3 cells in vitro [30]. Recently, a regulatory subpopulation of ILCs (called ILCregs) which can suppress the activation of ILC-1 and ILC-3 cells via the secretion of IL-10 and thus contribute to the resolution of innate intestinal inflammation, was identified [31]. This raises the question as to whether endogenous (and also exogenous) GCs may functionally potentiate these cells.

Hence, the idea that endogenous GCs effectively regulate ILC by suppressing ILC-derived cytokine production is gaining traction. This is probably essential to the prevention of early overshooting cytokine production upon infection that may otherwise cause (fatal) damage to the host. Whether GCs affect all cell types of ILC in a similar way remains to be investigated.

### 2.2. GC and T Cells

T cells are part of the adaptive immune system that provides the organism with the ability to recognize and respond to a wide variety of molecular structures (antigens) derived from various pathogens including viruses, bacteria, parasites and fungi. Depending on the nature of the pathogen and the signals provided by the antigen-presenting cells, helper CD4^+^ T cells produce the appropriate cytokines to effectively combat the pathogen involved [32]. One subset of helper T cells includes Foxp3^+^ regulatory T (Treg) cells, which counter-regulate ongoing immune responses and prevent autoimmunity [33]. T cells were the first immune cells studied in relation to regulation by endogenous GCs, showing that superantigen or polyclonal T cell activation appears lethal in a T cell-specific GR-deficient animal model. Protection was provided by treatment with a cyclooxygenase-2 inhibitor but not cytokine (IFN-γ) neutralization [17]. GR-deficiency in T cells (GR^Lck-Cre^ mice) was also fatal in an animal model of cecal ligation and puncture–induced polymicrobial septic death [18]. In an acute infection animal model, the inoculation of GR^Lck-Cre^ mice with the parasite *Toxoplasma gondii* created a hyperactive CD4^+^ T cell response and fatal immunopathology [19]. T cells, but not macrophages, are an important target for endogenous GC in order to limit clinical disease in experimentally induced autoimmune encephalomyelitis (EAE), a widely used animal model of multiple sclerosis (MS) [20]. Mice lacking the GR specifically in T cells (GR^Lck-Cre^ mice) developed EAE induced by myelin oligodendrocyte glycoprotein (MOG) peptide earlier than control mice. Furthermore, in contrast to the control mice, therapeutic treatment with dexamethasone was largely ineffective in these mice [20]. Interestingly, a recent study showed that EAE developed at a comparable rate over time in Treg cell-specific GR-deficient (GR^Foxp3-Cre^) mice and control mice, indicating that Treg cells are not the main target of endogenous GCs to restrain disease [25]. In contrast, GC treatment successfully suppressed the course of the disease but this appeared to be completely dependent on the presence of the GR in Treg cells. The results of these studies support the idea that endogenous GC attenuate clinical EAE by acting primarily on pro-inflammatory T cells, thereby limiting the production of cytokines such as IFN-γ, rather than potentiating regulatory T cell functions. Conversely, the failure of exogenous GC to inhibit disease symptoms in mice lacking GR signaling in Treg cells suggests that the therapeutic effect of GC (which is on top of the endogenous effect) is predominantly mediated via Treg cells.

Kim et al. also used Treg cell-specific GR-deficient mice to investigate the impact of GCs in a model of cockroach antigen-induced allergic airway inflammation and reported that GC treatment, analogous to the observations in the EAE model, was no longer able to suppress inflammation in the absence of GC signaling in Treg cells [25]. In contrast, Klaßen et al. reported that GR deletion in total T cells had no impact on the GC-mediated inhibition of OVA-induced allergic airway inflammation and that airway epithelial cells to be a major target for GC in this model [21]. Additional studies are required to clarify the relative contribution of T cell subsets as targets for GC-mediated suppression in these models of allergic airway inflammation. Evidence for a direct role of Treg cells as target cells for endogenous GC was reported by Rocamora-Reverte et al. GR-deficient Treg cells derived from GR^Foxp3-Cre^ mice were unable to prevent the induction of inflammatory bowel disease in a T cell transfer mouse model [26]. Inflammatory conditions allowed GR-deficient Treg cells to acquire Th1 cell-like characteristics as they expressed IFN-γ and failed to suppress pro-inflammatory CD4^+^ T cell expansion in situ. Hence, the results suggest that endogenous GC may stabilize both the phenotype and inhibitory function of Treg cells under inflammatory conditions. Along this line, a relative expansion of Treg cells (as compared to conventional CD4^+^ T cells) by steroid hormones (GC, progesterone) was reported in pregnant mice undergoing EAE. Both the expansion of Treg cells and pregnancy-induced protection from EAE were lost in pregnant GR^Lck-Cre^ mice, suggesting Treg cells are critical target cells for GC to induce tolerance to autoimmunity in pregnancy [34]. Moreover, the T cell-selective transgenic overexpression of GR reduced the basal T cell number, but Foxp3^+^ Treg cells were relatively resistant [35].

The results obtained in T cell-specific GR deficient animal models of inflammatory diseases thus show that endogenous GCs target T cells to limit clinical disease and improve survival. Consequently, one may expect enhanced GR signaling in T cells to improve inflammatory disease as well. Indeed, transgenic rats overexpressing a mutated GR with increased affinity towards GCs specifically in T cells displayed a delay in onset and strong reduction in the severity of EAE [36]. In contrast, endogenous GC did not affect disease intensity or progression in antigen-induced arthritis, an animal model exhibiting characteristic features of human rheumatoid arthritis. Yet, by using cell type-specific GR-deficient mice it could be demonstrated that exogenous GC treatment of inflammation in antigen-induced arthritis was effective when murine T cells, but not myeloid or B cells, expressed the GR [22].

T cells, and more specifically cytotoxic CD8^+^ T cells, are also targets for endogenous GC to control (acute) graft-versus-host disease (GvHD). When T cells from GR^Lck-Cre^ mice were transferred to induce acute GvHD in a fully MHC-mismatched recipient, clinical disease was strongly exacerbated compared to T cells taken from control GR^flox^ mice [24]. Exogenous GC treatment transiently improved acute GvHD only when recipients received GR^flox^ T cells, but did not have an effect on mortality. Using a single MHC class I-mismatch acute GvHD model in which disease was solely provoked by CD8^+^ T cells, the authors showed that the control of clinical symptoms by endogenous GC essentially depends on the suppression of cytotoxic CD8^+^ T cell function [24].

In summary, T cell-specific GR deficient animal models clearly demonstrate that T cells are essential target cells for endogenous GC to control infectious and inflammatory diseases. The inhibition of T cell-derived pro-inflammatory cytokine production is crucial for GC to protect animals against lethal immunopathology. In addition, it emerged that GC enhances, at least in some models, the function of physiologically suppressive Treg cells, thus contributing to host protection.

In other experimental settings, endogenous locally produced GCs act on T cells in a negative fashion which can be detrimental to the host. GCs produced in a tumor microenvironment (TME) of two transplantable murine cancer models promoted the dysfunction of tumor-infiltrating lymphocytes (TIL) [37]. More specifically, monocytes and macrophages in the TME were identified to produce de novo GC that promoted the differentiation of CD8^+^ TIL towards dysfunctional cells which produced less pro-inflammatory cytokines and up-regulated the expression of negative immune checkpoint molecules Tim-3 and PD-1. The development of these dysfunctional CD8^+^ T cells was prevented in mice carrying a conditional genetic ablation of the GR selectively in CD8^+^ T cells (GR^E8i-Cre^ mice) which also reduced tumor growth. Indeed, improved CD8^+^ T cell function and tumor growth control was similarly accomplished when mice were used in an experiment in which monocytes and macrophages were unable to synthesize GCs [37]. The effects of extra-adrenal, locally synthesized GC-regulating regional immune responses is a subject of increasing interest. While avoiding systemic effects, paracrine GC signaling has been shown to provide local immunosuppression, particularly at the epithelial barriers of the intestine, skin and lung (for review see [38]).

### 2.3. GC and Myeloid Cells

Granulocytes, macrophages and dendritic cells are subtypes of myeloid cells that are among the most important defenders against infection. They play a central role in local inflammation and tissue homeostasis and can initiate and sustain, or inhibit, T cell immunity [39,40]. Recently, myeloid cells were investigated to determine whether they are important target cells for GC under inflammatory settings. Endogenous GC did not modify disease progression in animal models for antigen-induced arthritis [22] and T cell-dependent allergic contact dermatitis [23] when mice lacked GR expression specifically in neutrophils and macrophages (GR^LysM-Cre^ mice). As indicated, the therapeutic benefit of glucocorticoid treatment in antigen-induced arthritis was dependent on GR-expressing T cells and was not diminished in GR^LysM-Cre^ animals [22]. In contrast, the opposite result was observed in contact dermatitis, whereby the expression of the GR in myeloid cells was required for exogenous GCs to inhibit allergic contact dermatitis, involving the suppression of high levels of leucocyte infiltration [23]. The deletion of the GR in T cells (or keratinocytes) did not change the efficacy of GCs to downregulate contact allergy although T cells are essential for the development of the allergic response. The impaired repression of inflammatory cytokines and chemokines such as IL-1β, monocyte chemoattractant protein-1 (MCP-1) and macrophage inflammatory protein-2 (MIP-2) were suggested to be responsible for the observed resistance to GC treatment in GR^LysM-Cre^ mice [23].

In a model of endotoxemia, mortality following a challenge with lipopolysaccharide (LPS) was much greater in mice that were GR-deficient in myeloid cells (GR^LysM-Cre^ mice) as compared to control mice, suggesting a role of endogenous GC [27]. Mortality was attributed to an increase in the release of circulating pro-inflammatory cytokines, such as TNF-α or IL-6, which are instrumental in inducing shock in this model [27]. In addition, a study employing mice with a CD11c-driven GR deletion specifically in dendritic cells (DCs; GR^CD11c-Cre^ mice), but not macrophages, identified IL-12 production by CD8^+^ DCs as a critical target for endogenous GC in order to protect from LPS-induced septic lethality [13]. Thus, both macrophages and DCs are critical cellular targets for the endogenous GC-mediated suppression of LPS-induced septic shock. Physiological GC action in myeloid cells is also essential to control the inflammatory response in dextran sulfate sodium (DSS)-induced colitis, a model for inflammatory bowel disease (IBD). In contrast to controls, GR^LysM-Cre^ mice failed to resolve colitis, produced persistently higher levels of IL-6 and did not recover from inflammation-induced weight loss [28]. Finally, GC mediate a crucial effect on myeloid cells in promoting tissue-repair upon myocardial infarction (the principal cell type was shown to be monocytes/macrophages). GR^LysM-Cre^ mice displayed reduced cardiac function, impaired wound healing and increased mortality after cardiac rupture [29].

In conclusion, myeloid cells are important target cells for both endogenous and exogenous GC to reduce or prevent damage to the host in many, but not all, mouse models of inflammation. The results obtained in these studies further support the idea that the effects of GC on immune homeostasis are cell type- and context-specific.

### 2.4. GC and B Cells

B cells are at the center of the humoral part of adaptive immunity and responsible for the production of antibodies directed against all classes of invasive pathogens [41]. B cells express high levels of GR and numerous studies indicate that both endogenous and exogenous GCs modulate B cell development, homeostasis and function [42]. In order to provide direct evidence for GC-related effects on B cell function, a few animal models harboring a specific deletion of the GR in the B cell lineage were generated. However, they did not reveal sizeable effects in inflammatory settings. In antigen-induced arthritis, GR^CD19-Cre^ mice showed a course of disease similar to that in control GR^flox^ mice, indicating endogenous GCs do not influence inflammation by B cells. As mentioned before, exogenous GC treatment efficiently suppressed clinical symptoms in this model by targeting T cells [22], but not B cells. Antigen-induced allergic airway inflammation is an experimental animal model in which B cell-derived antibodies are part of the pathophysiological process. Yet, neither endogenous nor exogenous GC were found to have any impact on the grade of airway hyperresponsiveness in GR^CD19-Cre^ mice, indicating GR expression in B cells is dispensable for the therapeutic efficacy of GC in this model [21]. Hence, despite high expression levels of the GR by B cells, a clear role for both endogenous and exogenous GC in B cell-driven immune responses has yet to be defined. Recently, an additional mouse model lacking GR expression in B cells was generated using the Mb1 promoter. While the humoral responses of these mice to immunizations with T-dependent and type 1 T-independent antigens were normal, a diminished response was documented upon immunization with a multivalent T-independent type 2 antigen. Interestingly, the homing of mature B cells to the bone marrow, but not to other lymphoid tissues, was impaired [43].

Taken together, animal models carrying a B cell-specific deletion of the GR demonstrate that endogenous GC do not target B cells in models of arthritis or allergic airway inflammation. Whether GC require B cells to attenuate inflammation or immune responses during infection in other disease models needs to be addressed in future studies.

### 2.5. GC and Other Immune Cells

Other cell types of the immune system are probably also affected by endogenous GCs, but only a few studies have been published to date. γδ T cells represent an unconventional T cell population and play a significant role at the interface between the innate and the adaptive immune systems. They colonize most peripheral tissues but are highly abundant in the gut mucosa as part of the intestinal intraepithelial lymphocyte (IEL) population that contributes to tissue homeostasis and the surveillance of infection [44]. While chronic restraint-stress decreased the number of γδ T cells in the epithelium of the small intestine, the blockade of the GR after treatment with the GR antagonist RU486 reversed this effect. Whether the stress-induced reduction in γδ T cells had any functional consequences for the epithelial barrier was not investigated [45].

Invariant natural killer T (iNKT) cells are innate-like T cells which recognize microbial and endogenous lipid antigens presented by CD1d molecules. They are powerful regulators of immune responses because of their capacity to produce large quantities of cytokines at an early stage [46]. The chronic restraint stress-elicited release of glucocorticoids produced iNKT cell dysregulation, which presented strongly reduced Th1 as well as Th2 responses upon the administration of the prototypic glycolipid ligand of iNKT cells, α-galactosylceramide [47]. In contrast, the same stress load failed to compromise the cytokine response in GR^Lck-Cre^ mice upon injection with α-galactosylceramide. Moreover, iNKT cells in stressed animals failed to trigger an antimetastatic response to B16 melanoma, an observation that was reversed by the pharmacological blockade of the GR. The authors also tested the effects of physical restraint on mucosa-associated invariant T (MAIT) cells, another population of innate-like T cells that respond to microbial riboflavin metabolites presented by MHC-related protein 1 (MR1) molecules [46]. Likewise, reduced Th1 and Th2 responses were observed in stressed animals upon the administration of the microbial metabolite 5-OP-RU [47]. The potential significance of these GC effects on MAIT cells for microbial infections and/or malignancies has not been investigated yet.

Clearly, more studies are required to investigate how endogenous GCs act on unconventional T cell lineages in inflammatory settings. So far, only chronic stress situations have been studied wherein the repeated release of GC negatively regulated the function of these cell types.

### 2.6. Endogenous GC and Microbial Clearance

It has been suggested that the (cell-specific) loss of responsiveness to endogenous GC and subsequent increased mortality in animal models may be caused by an inefficient immune response, leading to higher microbial burdens and pathologies directly caused by these microbes. However, this view is not supported by experiments directly addressing the in vivo quantification of microbes. While the *Trypanosoma cruzi* infection of mice induced a strong release of endogenous GC, the blockade of the GR accelerated death in these mice without affecting parasitemia [48]. Murine cytomegalovirus (MCMV) burden did not increase in adrenalectomized mice, which were more susceptible to virus-induced lethality. In parallel, circulating levels of cytokines IL-12, IFN-γ, TNF-α and IL-6 were increased and of these cytokines at least TNF-α was required for increased lethality in this setting [49]. Similarly, MCMV viral clearance was not compromised in mice lacking GR expression selectively in NK and ILC-1 cells and reduced survival in these mice appeared to be a consequence of increased IFN-γ production [12]. Investigating the parasitic disease malaria, Vandermosten et al. [50] reported that, in four different mouse models of malaria, endogenous GC conferred disease tolerance and protected against mortality. Indeed, adrenalectomy led to early severe hypoglycemia and excessive systemic and brain inflammation that was lethal, while parasitemia was not affected. The infection of mice carrying a T cell-specific deletion of the GR (GR^Lck-Cre^ mice) with the parasite *Toxoplasma gondii* produced hyperactive Th1 cell function and lethal immunopathology, although parasite numbers were comparable to control mice who survived infection [19]. In these GR^Lck-Cre^ mice, IFN-γ and TNF-α production by Th1 cells increased and Th1 cells drove mortality as the depletion of these cells increased survival. Interestingly, Th1 cells were shown to trigger the release of endogenous GCs, which subsequently inhibited cytokine production and the effector function of these Th1 cells by negative feedback [19]. Finally, influenza virus infection triggered a sustained increase in systemic GC and suppressed cytokine production in mice [51]. When the mice were co-infected with the pathogen *L. monocytogenes* 24 h after viral inoculation, the resulting hepatic bacterial burdens were much higher than those of mice infected with *L. monocytogenes* alone, an observation which was shown to be caused by the preceding virus-induced increase in serum GC. Strikingly, bacterial loads in co-infected adrenalectomized animals were much lower than in sham-operated mice and comparable to those in animals infected with *L. monocytogenes* alone. Importantly, the improved bacterial clearance in co-infected adrenalectomized animals was accompanied by a stronger inflammatory response and the mice succumbed to this co-infection (viral titers in co-infected adrenalectomized mice were similar to controls [51]).

Interestingly, human studies revealed that patients with Addison’s disease, i.e., suffering from primary adrenal insufficiency causing hypocortisolism, have an increased risk of death following infections [52,53]. However, since these patients are commonly treated with GC, it is presently not clear whether the enhanced risk of infection and death may be due to insufficient GC substitution, or adverse effects caused by GC replacement therapy itself [54].

Thus, evidence clearly shows that endogenous GCs are crucial for survival after infection with various microbes. The protective effect of these hormones is achieved by dampening the production of proinflammatory cytokines, rather than preventing the development of high microbial burdens (see Figure 2).

### 2.7. Concluding Remarks

Animal models deficient in GR expression specifically in various immune cell types (innate lymphoid, myeloid, T or B cells) have clearly shown that the ability of endogenous GCs to act as an essential negative regulator is cell-specific and depends on the experimental model used to study autoimmune and inflammatory diseases. Moreover, it has become clear that survival during infectious diseases critically depends on the GC-evoked suppression of the production of proinflammatory cytokines, and is not due to microbial burdens *per se*. These findings fit well with and emphasize the concept discussed already in 1984 by Munck and colleagues, stating that GCs protect an organism from the overactivity of its own immune response that may otherwise cause (fatal) damage to the host [55]. The experiments discussed in this review established that the protective effect of endogenous GC can be assigned to specific target cell types crucial for immunity against a given pathogen or for inducing inflammation. This creates a basis for the development of clinical therapies employing the cell-specific targeting of GC, which is currently underway (for further reading see [56,57] in this Special Issue). The ability of GC to simultaneously inhibit many key target genes involved in immune and inflammatory processes (Cain and Cidlowski, 2017), promises high therapeutic potency combined with a strong reduction in adverse side effects.

## Figures and Tables

**Figure 1 cells-11-02126-f001:**
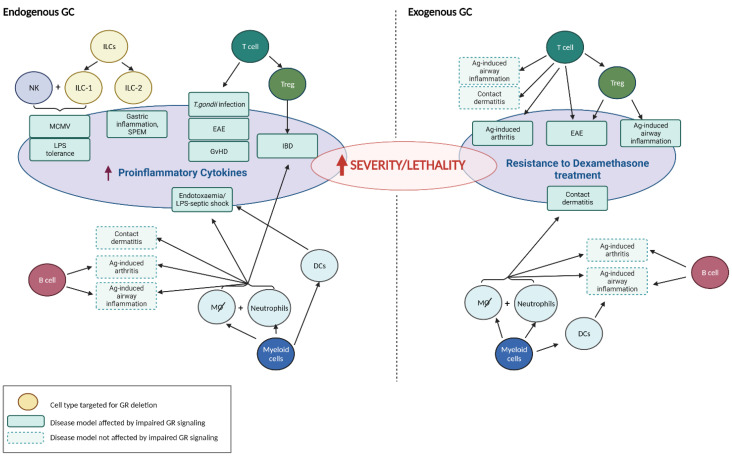
Immune cell-specific targeting by both endogenous and exogenous GC. Various immune cell types were experimentally targeted for GR deletion and then tested in animal models of infection and inflammation for actions by either endogenously released GC (**left panel**) or exogenous treatment with GC (**right panel**), or both. Disease models that are affected by impaired GR signaling in a certain immune cell type are indicated as green boxes in the central ovals. Animal models of disease not affected by impaired GR signaling in particular cells are depicted as boxes bordered by dashed lines. Cell type-specific regulation of inflammation and immunity by GC is seen, for example, in Ag-induced arthritis where T cells, but not B cells or myeloid cells, are targeted by GC (**right panel**). The red arrow means: ‘increased’ or ‘enhanced’. Created with BioRender.com (accessed on 24 April 2022).

**Figure 2 cells-11-02126-f002:**
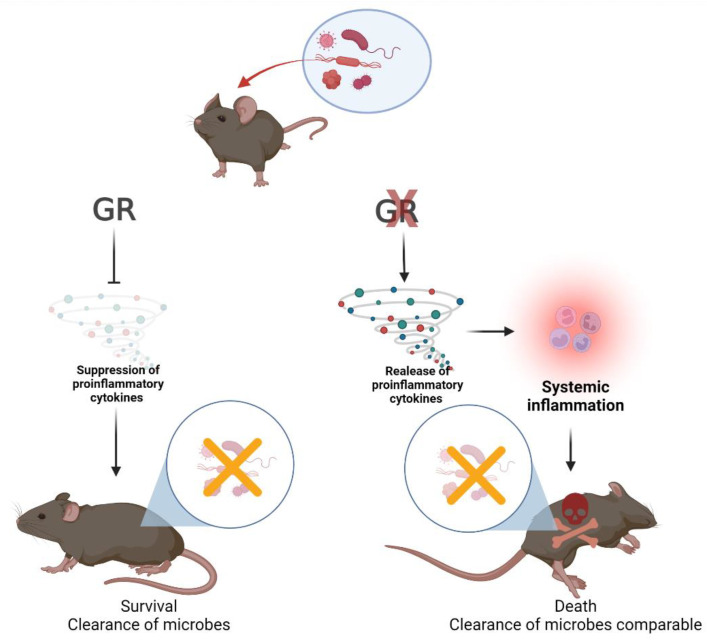
Cytokine hyperproduction, but not microbial load determines death or survival when GR signaling is impaired. Viruses, bacteria and parasites all induce effective immune responses by the host that are able to control these microbes. However, proinflammatory cytokine production must, in turn, be kept in check by endogenous GC-induced GR signaling (**left path**) because a lack of these hormones, or blocking GR signaling, may lead to cytokine hyperproduction, systemic inflammation and death (**right path**). So far, the cell type(s) targeted by GC have been identified in two experimental animal models: (i) a MCMV infection model where GC feedback on NK and ILC-1 cells [12]; and (ii) a *Toxoplasma gondii* infection model where GC feedback on T cells [19]. See Section 2.6 for details. Created with BioRender.com (accessed on 24 April 2022).

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
