# Peer review of "Cell-Specific Immune Regulation by Glucocorticoids in Murine Models of Infection and Inflammation"

_cells, 2022, doi:10.3390/cells11142126_

Round 1
Reviewer 1 Report
- This review by Rocamora-Reverte et. al., in general, provides an overview of the endogenous GCs in immune regulation, which is relevant and of interest to the field. However, in its current format, the article lacks a sufficient collection of studies in the field. Several reports on the roles of GCs in innate immunity and their immunosuppressive effects are not covered in the article. Moreover, authors have not provided sufficient updates on mechanistic insights into the endogenous GC mediated immune regulation.
- After addressing the following comments, I am confident the manuscript will be suitable for publication in Cells:
- The authors should elaborate on the current knowledge of GCs in circadian rhythms and their roles in both innate and adaptive immune modulation / regulation.
- I would suggest that the authors thoroughly go over the whole text and add the relevant citations at appropriate places. A few examples, please add the citations for lines 55-58, 58-61,116-118 and 119-122 and so on.
- Figure 1 should be created with the line thicknesses, symbols and text of sufficient size to ensure clarity. Please avoid using the greyscale symbols.
- In the legend of figure 1, please provide more explanatory text to understand the figure clearly without going into the main text.
- In section 2.2 (line 238-243), while highlighting the role of GR in DCs function, the authors have provided limited information in this line. Please also discuss the work by several other authors (e.g. Elftman MD et al. 2007, Chamorro S et. al. 2009 and Hodrea J et. al. 2012 and other recent papers).
- Please reformat the references as per the journal guidelines and avoid placing the citation in the middle of the sentence (e.g. line 151).
Author Response
We thank the reviewer for the time put into our manuscript and the detailed suggestions which clearly improved the quality of the manuscript. The focus of our review is to discuss the effects of endogenous (and where performed exogenous) glucocorticoids on specific immune cells in animal models of infection and inflammation. We consider it a short, focused review. That inherently leaves many aspects of glucocorticoid (immuno)biology uncovered. We agree with the reviewer that important aspects of glucocorticoid physiology in immunity should be mentioned. We, therefore, included two new references on glucocorticoids, circadian rhythms and immune regulation (Shimba et al., Immunity 2018; Shimba and Ikuta, Front. Immunol., 2020). In the introduction, we also included a few sentences on glucocorticoid mechanism of action and referred to Cain and Cidlowski, Nat. Rev. Immunol., 2017 for a detailed description.
We agree with the reviewer that references at the beginning of each section on glucocorticoids and specified immune cells were missing and have included the following new references in the revised manuscript:
We inserted a review on ILCs by Vivier et al., 2018 (lines 55-62). We inserted reviews on Thelper cell polarization by Tuzlak et al., 2021 (lines 116-120) and on regulatory T cells (Sakaguchi et al., 2020; lines 121-122). We inserted two references on myeloid cells and (regulation of) T cell immunity (Bassler et al., 2019; Yin et al., 2021; lines 214-216). We inserted a review on B cell responses by Cyster and Allen, 2019 (lines 257-258). Finally, we inserted references on γδ T cells (Ribot et al., 2021) and on iNKT and MAIT cells (Pellicci et al., 2020; lines 286-290).
We have modified Figure 1 and corrected the greyscale symbols.
As suggested, we have extended and clarified the text in the legend of Figure 1 (including an example of a glucocorticoid targeted cell (T cell).
In section 2.2 we indeed provided only limited information on the role of the glucocorticoid receptor in dendritic cell function. The reason is that we decided to include in vivo work only in this manuscript, which we applied to dendritic cells as well as to the other immune cell subsets discussed.
We also included a missing reference on glucocorticoid receptor-deficient B cells (Cain et al., 2020).
We apologize for not formatting the references according to the guidelines of Cells and have now done this in the revised paper. We also checked placements of references for their accuracy.
Reviewer 2 Report
In this manuscript, Lourdes Rocamora-Reverte et al. reviewed how glucocorticoids mediate their actions on immune responses that involve multiple cellular interactions between various immune cell subsets. The article emphasizes how these hormones regulate inflammation and immunity in a cell type- and context-specific manner, prevent infection-induced mortality by suppressing cytokine hyperproduction, and the lack of generation of microbial burdens by glucocorticoid deficiency or cell-specific resistance.
The subject of the article is of broad interest, and it is well written. Because the GR plays a key role in the mechanism of action glucocorticoids and it has recently been postulated as a key regulator for the variable biological response to cortisol according to the composition of its Hsp90-based heterocomplex, I feel that this part of the picture is missed here and should be addressed to present a more complete picture of the matter.
Author Response
We thank the reviewer for the time put into our manuscript and the suggestion to elaborate on the molecular mechanism(s) regulating the biological response to glucocorticoids. Our review focuses on the effects of endogenous (and where performed exogenous) glucocorticoids on specific immune cells in animal models of infection and inflammation. Many aspects of glucocorticoid (immuno)biology are, therefore, not covered. However, we agree with the reviewer that the mechanism of glucocorticoid action was not adequately described and a reference detailing this, was missing. To improve this in the revised manuscript, we now included a few sentences on glucocorticoid mechanism of action in the introduction and referred to Cain and Cidlowski, Nat. Rev. Immunol., 2017 for a detailed description.
Reviewer 3 Report
The manuscript reviewed the role of endogenoous glucocorticoids in the inflammatory or infectious immune respons of immune cells in animal models. It is interesting topic, however it is better to modify.
- There is only animal model data, so it is better to include human data such as ex vivo.
- If not, it is better to include the specification such as animal model in the title.
- In line 34-40, the paragraph is too long and hard to understand, so it may be seperated for more clear description.
- In line 377. the "A. Munck" should be changed to "Munck".
- In line 384-5, "The ability of GC to simultaneously inhibit many key target genes involved in immune and inflammatory processes" should be modified because there is no genetic data.
Author Response
We thank the reviewer for the time put into our manuscript and the proposed comments that improved our review and the title in particular. Thus, we changed the title to indicate that the review discusses work performed in murine animal models of infection and inflammation.
We have split the text in line 34-40 in three parts for clarity.
We have changed ‘A. Munck’ into ‘Munck’ (line 377).
In line 384 -385, a reference was missing and we now refer to Cain and Cidlowski, Nat. Rev. Immunol., 2017.
Round 2
Reviewer 1 Report
This revised version is much improved and almost acceptable for publication. I have only one comment left to address, which refers to figure 2: As it is, the figure is somewhat 'nondescript' and oversimplified. Which cell types (which are discussed in the text!) are being referred to? What is the take home message here, are there specific molecules or mechanisms which would be relevant here? Please include more scientific detail and content in Figure 2.
Author Response
We thank the reviewer for the appreciation of our revised manuscript. Regarding figure 2, animal models in which the cell type(s) targeted by GC have been identified to date, are:
- a MCMV infection model where GC feedback on NK and ILC-1 cells. Survival of mice in this infection model appeared to depend on GC-induced suppression of pathological IFN-γ production by splenic NK cells (Quatrini et al. Nat Immunol 2018, 19, 954-962).
- a Toxoplasma gondii infection model where GC feedback on T cells. Mortality of T cell-specific GR-deficient mice in the Toxoplasma gondii infection model strictly depended on hyperactive Th1 cell function and associated immunopathology (Kugler et al, J. Exp. Med 2013, 210, 1919-1927).
Both animal models are described in the main text of section 2.6. In the legend of figure 2, we now mention these animal models and refer to the main text for details. Our intention with figure 2 is to depict a simplified scheme of what happens in the context of microbial clearance in all animal models we described in this review.
Reviewer 2 Report
The first version of this article had been positively evaluated by all reviewers, such that minor observations were made to improve the manuscript. Surprisingly, the authors have not made a significant effort to fulfill those expectations. Thus, the suggestion to address the molecular mechanism of activation of GR (roles of Hsp90, FKBPs, nuclear translocation, receptor dimmerization, etc.) was limited to 6-7 lines with limited information and showing a reference that is not related to the matter. Anyway, if the authors are happy with their "improved" article, I see no reason to impair its publication.
Author Response
We thank the reviewer for the effort put in our manuscript and appreciate the willingness to let us choose and discuss the topics on in vivo cell-specific immune regulation by glucocorticoids without restrictions. Clearly, we agree with the reviewer that the mechanisms underlying GR activation are highly important and should also be addressed in the in vivo studies described in the manuscript. However, we think a separate review would be required to properly address (levels, composition and function of) the chaperone/co-chaperone/GR complex, which is a vast topic.
Reviewer 3 Report
The authors respond adquately.
Author Response
We thank and appreciate the reviewer for accepting our revised manuscript in its current form.